Medical Imaging with Deep Learning 2024

# SDDA-MAE: Self-distillation enhanced Dual Attention Masked Autoencoder for Small-scale Medical Image Datasets

**Yunze Wang**[*1]                                          Yunze.Wang19@student.xjtlu.edu.cn
**Silin Chen**[*2]                                                       19271205@bjtu.edu.cn
**Tianyang Wang**[1]                                Tianyang.Wang21@student.xjtlu.edu.cn
**Jingxin Liu**[1]                                                Jingxin.Liu@xjtlu.edu.cn
[1] *Xi'an Jiaotong - Liverpool University*
[2] *Beijing Jiaotong University*

## Abstract

Masked Autoencoder (MAE) has shown promise as a self-supervised learning method in natural images. However, its application in medical imaging is limited by data scarcity. To alleviate this challenge, we propose **SDDA-MAE**, a method for direct pre-training and fine-tuning on targeted datasets without the requirement of self-supervised pre-training on an extra large dataset. The Dual Attention Transformer (DAT) serves as the backbone for enhanced spatial and channel-wise image representation. During the pre-training stage, we employ Self-distillation (SD) to transfer knowledge from the decoder, containing global information, to the encoder, which holds local information, improving weight initialization for downstream tasks. Experimental results demonstrate our method outperforms numerous self-supervised and supervised state-of-the-art (SOTA) methods in tasks like medical image segmentation and classification, even without pre-training on larger upstream datasets.

**Keywords:** MAE, Self-Distillation, Transformer, Pre-training, Small-scale Datasets

## 1. Introduction

Recently, Masked Autoencoder (MAE) has shown promising performance in self-supervised representation learning for natural image processing. However, its advancement in medical image analysis is hindered by the absence of large-scale datasets.

To make MAE adapt to small-scale medical image datasets, we introduce **SDDA-MAE**, a **S**elf-**D**istillation enhanced **M**aksed **A**uto**E**ncoder featuring a **D**ual **A**ttention Transformer backbone, as illustrated in Figure 1. The differences between SDDA-MAE and MAE (He et al., 2022) mainly lie in two aspects. Firstly, our model utilizes a redesigned backbone called Dual Attention Transformer (DAT), based on the architecture of DAE-Former (Azad et al., 2023), which efficiently processes the entire spatial dimension of input features and captures channel context more effectively compared to ViT (Dosovitskiy et al., 2020). Secondly, we incorporate Self-distillation (SD) (Zhang et al., 2019) during the pre-training stage, where the encoder acts as the student network and the decoder as the teacher network. This process minimizes the discrepancy between the output distributions of the two networks, encouraging the encoder to replicate the global features observed by the decoder. By integrating these two enhancements, our pre-training procedure enhances the feature representation learning capability, reducing the need for extensive pre-training datasets. Taking advantage of the consistent model architecture during both the pre-training and fine-tuning stages, we transfer the weights of both the encoder and decoder modules for downstream tasks, rather than solely transferring the encoder module weights. This approach is expected to yield a more optimal initial parameter space for downstream tasks, consequently enhancing performance.

---

[*] Contributed equally

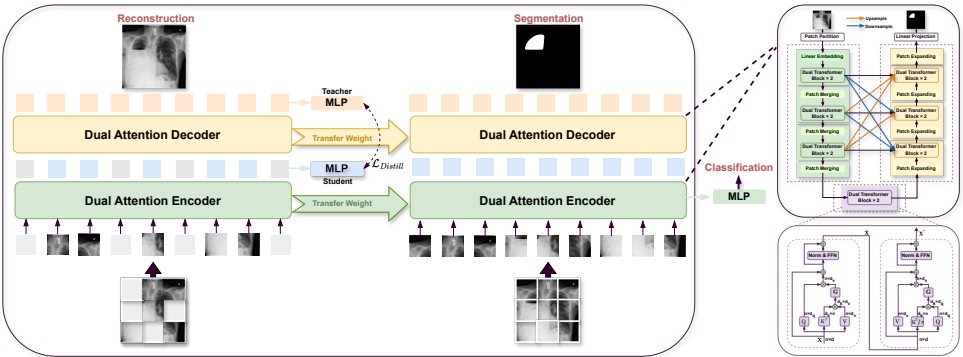

Figure 1: The workflow of proposed SDDA-MAE for small-scale medical image datasets.

## 2. Methods

The workflow of our proposed SDDA-MAE (as shown in Figure 1) proceeds as follows: in the pre-training stage, we retain the mask tokens but replace them with a shared learnable vector as the input of the encoder. These tokens are then processed through successive DAT blocks and patch merging blocks for feature extraction. It is worth noting that we utilize a masking strategy inspired by Swin-MAE (Dai et al., 2023) to prevent the model from learning shortcut solutions. Subsequently, the resulting feature representations undergo further processing via successive DAT blocks and patch expanding blocks to reconstruct the image in the decoder. In addition, we derive $V_{encoder}$ and $V_{decoder}$ by passing the outputs of the encoder and decoder through a single-layer MLP, followed by employing cross-entropy loss to minimize the difference between the distributions of the two vectors. The overall loss function $\mathcal{L}_{unsup}$ for the pre-training stage is expressed as below:

$$\mathcal{L}_{unsup} = \mathcal{L}_{MSE}(Y_{pred}, Y) + \mathcal{L}_{CE}(V_{encoder}, V_{decoder}),$$

where $Y_{pred}$ refers to the prediction of the masked patch, $Y$ refers to the ground truth, $V_{encoder}$ and $V_{decoder}$ refers to the output of the corresponding MLP layer respectively. During the fine-tuning stage, we transfer the weights of the encoder and decoder obtained in the pre-training stage and fine-tune the weights based on the following supervised loss $\mathcal{L}_{sup}$:

$$\mathcal{L}_{sup} = 0.5 \times (\mathcal{L}_{Dice}(S_{pred}, S) + \mathcal{L}_{CE}(S_{pred}, S)) + 0.5 \times (\mathcal{L}_{CE}(C_{pred}, C) + \mathcal{L}_{FL}(C_{pred}, C)),$$

where $S_{pred}$ and $C_{pred}$ refer to the prediction of segmentation and classification task respectively, and $\mathcal{L}_{FL}$ refers to the focal loss (Lin et al., 2017). Different from the pre-training stage, to enhance the segmentation performance of the model, we introduce full-scale skip connection operations between the encoder and decoder of SDDA-MAE.

## 3. Experiments and Conclusion

**Dataset.** Two datasets are used for evaluating our proposed model. The SIIM-ACR Pneumothorax Segmentation dataset (Anna Zawacki, 2019) comprises 12,089 annotated chest X-ray images, following the official data split as provided. Meanwhile, the BUSI Breast Cancer Segmentation dataset (Al-Dhabyani et al., 2020) contains 780 ultrasound images, with 80% of the data allocated to a training set and the remaining 20% designated for testing. Moreover, 10% of the two training sets are used for validation.
**Setting.** All images in both datasets are resized to $512 \times 512$ and employ random flip and

crop for data augmentation. During the pre-training stage, the initial learning rate is set to $2e^{-4}$, with a weight decay of 0.05, and a cosine schedule with warm-ups is employed. The number of pre-training epochs for SIIM-ACR and BUSI are 400 and 200, respectively, with batch sizes of 24 and 12. During the fine-tuning stage, their learning rates are set to $1.5e^{-3}$ and 0.01. The number of epochs is 100 and 50 with batch sizes of 24 and 12, respectively.

**Results.** We reported the detailed performances of SDDA-MAE and other self-supervised and supervised algorithms on two multi-task medical image datasets. As demonstrated in Tables 1, 2, 3 and 4, our model significantly outperformed other state-of-the-art (SOTA) self-supervised learning methods on both tasks. In addition, we compared our model with two SOTA supervised learning methods based on ImageNet pre-training. The results indicate that despite employing a significantly smaller pre-training dataset in comparison to ImageNet, our method marginally outperforms other supervised learning methods. Furthermore, as shown in Figures 2 and 3, we explored the impact of different masking ratios on downstream task performance, with a masking ratio of 60% yielding the best results.

Table 1: Segmentation performances on SIIM-ACR.

| Method | Dice(%) ↑ | Jaccard(%) ↑ | HD$_{95}$↓ | ASD↓ |
|---|---|---|---|---|
| *Self-supervised methods* | | | | |
| MAE(He et al., 2022) | 82.76 | 73.94 | 14.98 | 4.88 |
| MoCov3(Chen et al., 2021) | 81.98 | 73.12 | 15.32 | 5.21 |
| *Supervised methods* | | | | |
| UNet++(Zhou et al., 2019) | 84.12 | 78.32 | 13.23 | 4.02 |
| Swin-UNet(Cao et al., 2022) | 84.49 | 78.91 | 12.92 | 4.13 |
| *Ablation studies* | | | | |
| Only Encoder (DAT) | 83.07 | 74.23 | 14.67 | 4.54 |
| Only Encoder (DAT & SD) | 83.55 | 74.80 | 14.18 | 4.32 |
| Encoder & Decoder (DAT) | 83.41 | 74.39 | 14.58 | 4.41 |
| **Encoder & Decoder (DAT & SD)** | **85.75** | **80.31** | **10.87** | **3.02** |

Table 2: Classification performances on SIIM-ACR.

| Method | ACC(%) ↑ | PRE(%) ↑ | REC(%) ↑ |
|---|---|---|---|
| *Self-supervised methods* | | | |
| MAE(He et al., 2022) | 90.21 | 79.04 | 88.51 |
| MoCov3(Chen et al., 2021) | 89.04 | 77.83 | 87.84 |
| *Supervised methods* | | | |
| ViT-B/16(Dosovitskiy et al., 2020) | 93.23 | 81.52 | 92.12 |
| ResNet50(He et al., 2016) | 92.81 | 80.32 | 90.90 |
| *Ablation studies* | | | |
| Only Encoder (DAT) | 90.75 | 80.73 | 90.01 |
| Only Encoder (DAT & SD) | 92.96 | 81.13 | 91.58 |
| Encoder & Decoder (DAT) | 91.64 | 80.01 | 90.42 |
| **Encoder & Decoder (DAT & SD)** | **94.64** | **84.21** | **92.42** |

Table 3: Segmentation performances on BUSI.

| Method | Dice(%) ↑ | Jaccard(%) ↑ | HD$_{95}$↓ | ASD↓ |
|---|---|---|---|---|
| *Self-supervised methods* | | | | |
| MAE(He et al., 2022) | 79.91 | 73.06 | 15.98 | 4.93 |
| MoCov3(Chen et al., 2021) | 78.45 | 73.00 | 16.23 | 4.99 |
| *Supervised methods* | | | | |
| UNet++(Zhou et al., 2019) | 82.15 | 75.57 | 14.96 | 4.35 |
| Swin-UNet(Cao et al., 2022) | 83.49 | 77.39 | 14.48 | 4.02 |
| *Ablation studies* | | | | |
| Only Encoder (DAT) | 81.78 | 74.88 | 15.23 | 4.54 |
| Only Encoder (DAT & SD) | 82.45 | 76.03 | 14.98 | 4.20 |
| Encoder & Decoder (DAT) | 82.03 | 75.22 | 15.09 | 4.41 |
| **Encoder & Decoder (DAT & SD)** | **84.12** | **77.83** | **14.32** | **3.93** |

Table 4: Classification performances on BUSI.

| Method | ACC(%) ↑ | PRE(%) ↑ | REC(%) ↑ |
|---|---|---|---|
| *Self-supervised methods* | | | |
| MAE(He et al., 2022) | 89.31 | 89.12 | 88.90 |
| MoCov3(Chen et al., 2021) | 88.96 | 89.10 | 88.67 |
| *Supervised methods* | | | |
| ViT-B/16(Dosovitskiy et al., 2020) | 90.29 | 90.45 | 90.34 |
| ResNet50(He et al., 2016) | 91.23 | 91.33 | 91.37 |
| *Ablation studies* | | | |
| Only Encoder (DAT) | 90.03 | 90.41 | 90.34 |
| Only Encoder (DAT & SD) | 90.67 | 90.88 | 89.70 |
| Encoder & Decoder (DAT) | 90.34 | 90.60 | 89.48 |
| **Encoder & Decoder (DAT & SD)** | **92.53** | **92.32** | **92.39** |

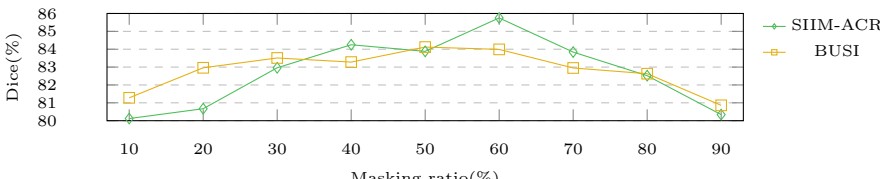

Figure 2: Segmentation performances using different masking ratios. **(Our Best Model)**

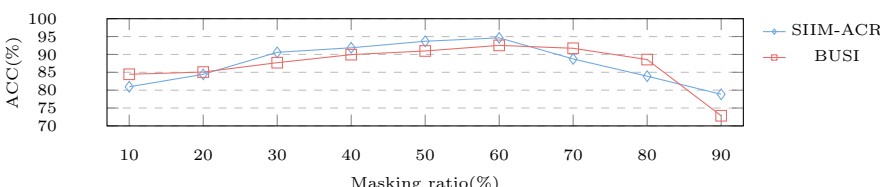

Figure 3: Classification performances using different masking ratios. **(Our Best Model)**

**Conclusion.** In this study, we introduce **SDDA-MAE**, a two-stage self-supervised framework aimed at fully extracting meaningful semantics from small-scale medical image datasets to enhance downstream task performance. Through comprehensive experiments along with ablation studies, we demonstrate the effectiveness and applicability of our proposed model.

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
