# OpenReview forum: "SDDA-MAE: Self-distillation enhanced Dual Attention Masked Autoencoder for Small-scale Medical Image Datasets"
_MIDL.io/2024/Short_Papers — MIDL 2024 Short Papers_

### Official Review · Reviewer_STW4 · 2024-04-24

**Confidence:** 5
**Final Rating:** 4

**Review:**

Strength:
This paper proposed a two-stage self-supervised framework aimed at fully extracting meaningful semantics from small-scale medical image datasets to enhance downstream task performance. Experimental results demonstrated better performance than the other methods.

Weakness:
It would be interesting to see more experimental comparison with recent pre-training methods on larger scale datasets, such as following:
CXR-CLIP: Toward Large Scale Chest X-ray Language-Image Pre-training. MICCAI 2023.

Summary:
Overall, this is an interesting paper aiming to work on small-scale medical datasets.

---

### Decision · Program_Chairs · 2024-04-26

Accept